# GENERATIVE TRAFFIC SIMULATIONS

## ABSTRACT

Controlling urban traffic is an emerging challenge for modern cities. In this work, we present a cascaded AI system that integrates a deep neural network with a large language model to generate and simulate both routine and edge-case traffic scenarios. The proposed system analyzes real-time congestion patterns and generates optimized traffic signal plans, adjusting signal timing based on vehicle flow direction and phase-level congestion to minimize delays and enhance throughput. By combining real-world traffic data with synthetically generated scenes, the approach enhances travel efficiency and minimizes wait times. To our knowledge, this is the first system that combines real-life video feeds, vehicle tracking and a large language model based code generator to create synthetic traffic scenarios and optimize traffic signal plans in SUMO.

## 1 INTRODUCTION

Simulating traffic scenarios is crucial to understand and optimize urban traffic systems. Tools like **SUMO**, **VISSIM**, and **CityFlow** are commonly used for simulating traffic scenarios (Li et al., 2024) and employing reinforcement learning in traffic signal optimization (Wei et al., 2018), (Oroojlooy et al., 2020) while simulators like **CARLA** and **LGSVL** are being used for autonomous driving research in modern cities (Dosovitskiy et al., 2017),(Rong et al., 2020). Researchers usually use CARLA and LGSVL to simulate both normal and adverse scenarios (Wei et al., 2024) and then evaluate how autonomous vehicles react under risky driving conditions (Zhang et al., 2024) (Peng et al., 2025). Each of these tools has its own strengths. Among these, **SUMO** (Simulation of Urban MObility) stands out for its microscopic modeling of traffic dynamics, compatibility with real-time data, and open-source accessibility. It is widely used for urban traffic planning, testing signal control strategies, and urban mobility research (Oroojlooy et al., 2020). However, one of the main challenges with such microscopic simulation tools is that one cannot use this software without having any prior knowledge of traffic simulations and traffic-related details. One needs to know how to define road geometry, traffic logic, vehicle behaviors, vehicle types, routes for each vehicle, etc. (Algherbal & Ratrout, 2025).

SUMO allows users to create and simulate complex road networks, define traffic signal logic, test vehicle behaviors, and analyze traffic patterns under diverse edge-case scenarios. Recent studies have shown to integrate large language models into SUMO simulators for studying traffic scenarios, congestion and dynamic rerouting. Leveraging large language models ChatSumo (Li et al., 2024) and Sumo-MCP (Ye et al., 2025) automate traffic simulation through network generation, traffic flow generation.

Congestion in intersections poses a significant challenge in traffic management. Optimization of Traffic Signal Logic (TSC) has been proven to be an effective solution in traffic systems. SUMO is an open-source platform for modeling and testing various TSC plans, enabling researchers to observe how optimized signal plans with timing adjustments and real-time interventions can reduce delays and improve traffic flows. In SUMO, there are various methods for handling traffic signal control. They can be categorized into three groups: traditional rule-based methods, reinforcement learning (RL) methods, and large language model (LLM)-based approaches. Traditional rule-based methods, such as Fixed-Time Control, Webster's Method (Ali et al., 2021), Self-Organizing Traffic Light Control (SOTL) (Cools et al., 2008), and Max-Pressure Control (Zoabi & Haddad, 2024), use predefined logic or an algorithm to determine or switch signal phases and their durations. However, they cannot perform well in all the diverse congestion scenarios. RL-based TSC is an approach where the model studies traffic signal control actions by experimenting with the traffic environment,

learns to optimize its control actions after getting feedback dynamically in the form of a reward function (Wei et al., 2019). However, like traditional methods, RL-based methods are not without limitations as well. Although it performs well in dynamic situations, it is unable to perform in unforeseen situations. It cannot handle situations like a broken car in the intersection, a road blockage due to harsh weather, or an emergency vehicle. So Traditional and RL based methods face overfitting issues due to being trained on specific traffic patterns.

To overcome situations like these, traffic signal control researchers nowadays are shifting their focus to LLM-based approaches. LLM-based approaches can handle any complex traffic scenarios that require critical human reasoning, which is again difficult to achieve with traditional rule-based or RL methods. Unlike pre-trained RL models that depend mostly on environment-specific training data, LLMs can generalize across diverse scenarios and respond accordingly after it has reasoned over high-level traffic patterns. Some significant recent works, such as *LLM-Assisted Light* Wang et al. (2024a), *CollMLight* Yuan et al. (2025), have made remarkable use of LLMs in managing traffic congestion by implementing context-aware strategies. Inspired by recent LLM-based works, we study a different but complementary problem: leveraging LLMs for both scenario generation and offline signal plan generation at a single four-way intersection. Specifically, our contributions are: (a) SUMO-compatible scenario generation from natural language prompts, enabling both normal and edge-case conditions; and (b) offline, congestion-aware signal plan generation, where simulation is first run, congestion is assessed, and then the LLM generates static timing plans for a subsequent run.

This framing positions our system as an offline code-generation pipeline, distinct from optimization-oriented approaches. Our approach consists of the following key components:

- **Real-time traffic data collection:** We Collect live traffic video feeds from a four-way urban intersection and apply deep learning-based detection and tracking algorithms to extract vehicle trajectories. These trajectories are aggregated into phase-level occupancy signals.

- **LLM driven Traffic scenario generation:** Unlike prior LLM-TSC papers which only focused on signal control, we use LLMs to create both everyday and edge-case traffic scenarios. Then we simulate these scenarios in SUMO using extracted traffic data.

- **Injecting congestion events into traffic data:** We introduce synthetic congestion events, such as the sudden arrival of a large number of vehicles into normal traffic data extracted from a real-time feed. Then the performance of the LLM to handle the congestion resulted from these events is evaluated.

- **LLM-guided offline traffic signal plans:** LLM is prompted to generate Python code that updates traffic signal logic files based on which phases are maximum in occupancy calculation. It creates different plans for different phase groups. Our LLM generated plans are offline and static which are then evaluated in the next simulation run, rather than optimized online during runtime as in prior TSC work.

## 2 Literature Review

### 2.1 Simulation in Traffic Research

Traffic simulators are becoming increasingly popular in the urban traffic research community. Simulators such as **CARLA** and **LGSVL** (Zhang et al., 2024) are widely used in autonomous driving research for tasks such as crash or collision prevention, safety protocol validation, and sensor robustness testing . On the other hand, microscopic traffic simulators like **SUMO** and **CityFlow** (Yuan et al., 2025) are commonly used to address traffic congestion problems and optimize traffic signal control. Recent research works like **ChatScene** (Zhang et al., 2024) leverage LLMs to generate adversarial, safety-critical scenarios in 3D environments from natural language prompts and help in reducing collisions. It reduces the collision rate to 9%. **LD-Scene** (Peng et al., 2025) uses LLM with Latent Diffusion Models (LDMs) to generate diverse and adversarial driving scenarios from natural language prompts. The frameworks also include Chain-of-Thought code generator and debugger to increase precision and ensure robustness. Other frameworks such as **ChatSim** (Wei et al., 2024) utilize LLMs to run editable, photo-realistic 3D driving scene simulations from natural language prompts. This system focuses on simulating realistic scene videos. **Traffic scene generation from**

**natural language description for autonomous vehicles with LLM** (Ruan et al., 2024) uses a text-to-traffic scene framework that employs an LLM to generate both standard and critical traffic scenarios for the CARLA simulator based on natural language descriptions. **ChatSUMO** (Li et al., 2024) also uses LLM to generate traffic simulations in SUMO from natural language prompts. The system configures road networks after fetching map data from OpenStreetMap when LLM is given the command to show traffic in any particular city, and then runs SUMO simulations.

## 2.2 SIGNAL CONTROL IN TRAFFIC RESEARCH

Apart from simulation, TSC is another major focus in urban traffic research. Microscopic simulators are commonly employed to reproduce congestion scenarios and test different solutions under complex environments.

**Traditional Rule-Based Method:** Webster (Wei et al., 2019) method computes the optimal cycle length and splitting of traffic signal phases at isolated intersections based on traffic flows. Self-Organizing Traffic Light Control (SOTL) (Cools et al., 2008) follows predefined logic to decide whether to switch to another phase or continue the phase based on local vehicle counts. There is a counter that calculates how many vehicles enter the red light zone; if the threshold is crossed, then it decides to switch. Max-Pressure Control (Varaiya, 2013) decides which phase to activate based on the maximum difference between approaching and exiting queue lengths in the intersections to improve the overall traffic travel time. **RL Based Method:** Presslight (Krajzewicz et al., 2002) combines the Reinforcement method with the Max-Pressure theory. The intersection max pressure is key input for the reward function in RL. IntelliLight (Wei et al., 2018) uses the number of vehicles, average waiting time, and lane-wise queue length details. The reward function is mapped to how frequently traffic signals are being changed. UniTSA (Wang et al., 2024b) manages TSC in a Vehicle-to-Everything (V2X) environment. It uses a junction matrix to represent standard intersection and makes use of traffic state augmentation based on intersection symmetry. **LLM Based Method:** LLM Assisted Light (Wang et al., 2024a) optimizes traffic signals dynamically based on maximum occupancies in each phase. For example, when an emergency vehicle suddenly arrives, even if other lanes are slightly congested, that phase will be activated. CoLLMlight (Yuan et al., 2025) optimizes traffic signals dynamically in **Cityflow** simulator. Still, it handles traffic signal control not at a single intersection level, but city-wide.

Inspired by the success and performance of existing works, we have implemented an LLM within the SUMO traffic simulator to handle both normal and abnormal traffic scenarios. Our method is capable of generating routine traffic flows and scenes, as well as simulating congested scenarios. Using the reasoning capabilities of the LLM, the system effectively mitigates congestion issues and manages special traffic situations that often do not fall into the capacities of traditional RL-based methods or even other existing LLM approaches. Unlike other LLM-based methods, our approach does not decide which phase to switch or activate; It rather changes traffic signal plans entirely by generating the simulation code.

## 3 METHODOLOGY

### 3.1 OVERVIEW

Our methodology makes use of an LLaMA 3.2 1B model, fine-tuned using QLoRA framework, to generate and optimize traffic scenarios at a four-way intersection in the SUMO simulator. The LLM generates Python code to create both everyday and edge case traffic scenarios and then produces optimized traffic signal plans to reduce congested traffic movements and improve traffic flow.

### 3.2 INTERSECTION MODELING

To replicate realistic traffic dynamics, we model twelve directional movements corresponding to straight-through and left-turn flows from the four arms of the intersection (North, South, East, West). These movements are categorized into eight *lane groups*:

- **NT, ST, ET, WT** — straight-through movements from north, south, east, and west.
- **N-L, S-L, E-L, W-L** — left-turn lanes in north, south, east, and west.

For traffic scenario generation and signal control, these lane groups are aggregated into four *phase groups*, each corresponding to one controllable signal phase:

- **NS** — North–South straight movements (NT, ST)

- **NS-L** — North–South left turns (N-L, S-L)

- **EW** — East–West straight movements (ET, WT)

- **EW-L** — East–West left turns (E-L, W-L)

### 3.3 CONGESTION MEASUREMENT

At each simulation timestep $t$, SUMO reports the instantaneous occupancy of every lane $l$, denoted $occ_l(t) \in [0, 1]$, where $occ_l(t)$ represents the fraction of lane length occupied by vehicles. We average the occupancies of those lanes.

**Lane-group occupancy** The occupancy of a lane group $g$ at time $t$ is given by

$$O_g(t) = \frac{1}{|\mathcal{L}_g|} \sum_{l \in \mathcal{L}_g} occ_l(t), \tag{1}$$

where $\mathcal{L}_g$ is the set of lanes in group $g$.

**Time-averaged lane-group occupancy** Across a simulation of $T$ timesteps,

$$\bar{O}_g = \frac{1}{T} \sum_{t=1}^{T} O_g(t). \tag{2}$$

**Phase-level occupancy** Each phase $p$ consists of multiple lane groups. Its maximum average occupancy is defined as

$$\bar{O}_p = \max_{g \in \mathcal{G}_p} \bar{O}_g, \tag{3}$$

where $\mathcal{G}_p$ is the set of lane groups belonging to phase $p$. The most congested phase is then selected as

$$p^* = \arg \max_{p \in \mathcal{G}_{\text{phases}}} \bar{O}_p. \tag{4}$$

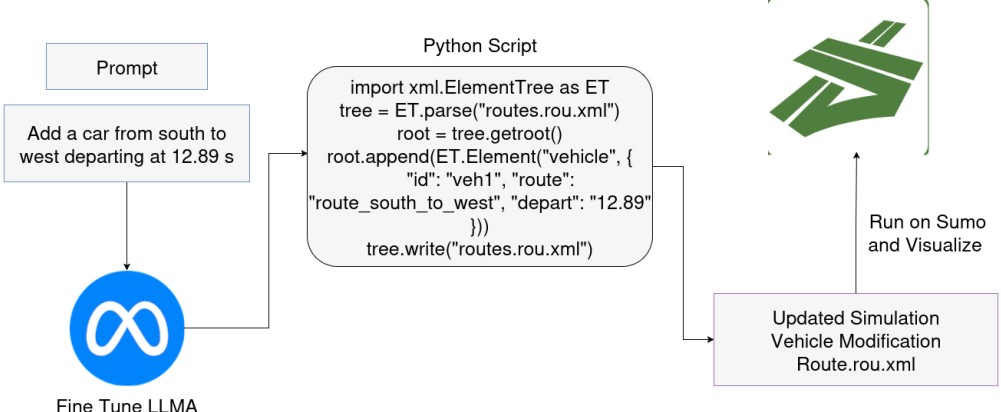

Figure 1: Standard Traffic Scene Generation

## 3.4 LLM-DRIVEN SCENARIO GENERATION

LLM as a code generator, when a prompt describing a traffic condition is given, it outputs Python scripts that directly write SUMO-compatible XML files. This allows us to create traffic scenarios without manually coding XML. We design three unique scenes:

**Scene 1: Normal Traffic Flow** The LLM is asked to generate code for normal traffic demand from high-level description (e.g., add a vehicle from north to south) and then generates Python code that automatically updates the route.rou.xml file, specifying unique vehicle ID, departure time, and directional route movement (e.g., north-to-south). Figure 1 illustrates a typical example of such a normal traffic flow, where the LLM-generated script produces routine intersection movements through XML files.

**Scene 2: Sudden Vehicle Breakdown** In this scenario, the LLM is instructed to place a broken vehicle near the intersection that breaks down while it is taking a left turn. Breakdown can occur in any of the four directions: north, south, east, or west, which can create localized congestion, and we test the robustness of the system to disruptions. For example, given a prompt like A car left at 0.28 s stalls in the west left-turn pocket, backing up traffic, LLM generates python code which updates route.rou.xml file by inserting a special vehicle with a predefined stop position on the west left turn lane at the given time, causing a stall.

**Scene 3: Heavy or Medium Traffic Flow** High or medium density traffic conditions can be created by giving a command (e.g., Generate a heavy scenario with 231 cars in the network) to LLM to produce python code to generate the route.rou.xml file containing total 231 unique vehicle IDs, randomized departure times, and predefined directional routes, sorted by departure time.

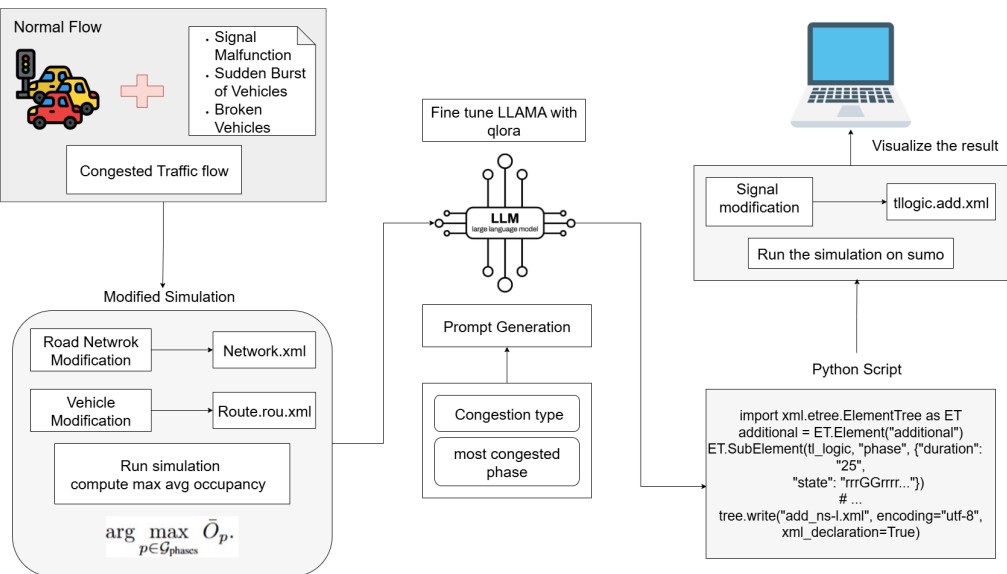

Figure 2: Proposed traffic signal optimization.

### 3.4.1 SCENE CUSTOMIZATION CATEGORIES

To test the LLM's performance, we prepared three congestion scenarios. They were scripted manually.

**Traffic-Signal Malfunction** One or more approaches remain on red light for an abnormally long time, which prevents vehicles from clearing the intersection, thus causing a huge queue of vehicles and congestion.

**Sudden Burst of Vehicles** A large, short-lived burst of traffic is added along a single direction (e.g., South→North) to test event-driven surges such as before a sports game or during morning office hours. Each of the twelve directions is are tested individually, rather than all at once.

### 3.5 OFFLINE CONGESTION-AWARE SIGNAL PLAN SYNTHESIS

First, we introduce synthetic congestion events—such as signal malfunctions, sudden bursts of vehicles, and stalled vehicles—into otherwise normal traffic flows. These events are encoded by modifying the signal timing in network.xml and the vehicle definitions in route.rou.xml. The congested scenario is then executed in SUMO, where we compute the average occupancy per phase and identify the phase with the maximum occupancy percentage, along with the corresponding type of congestion. After the simulation, the LLM is prompted with this information (congestion type, most congested phase, and its occupancy percentage) to generate optimized traffic signal plans. The LLM outputs Python code that produces updated, phase-based signal plans in tllogic.add.xml. Finally, the scenario is rerun in SUMO to compare traffic performance metrics before and after the application of the LLM-generated plans. The objective of these optimizations is to improve overall traffic flow and reduce the average delay at the intersection. Figure 2 and Algorithm 1 show how LLM handles these three congestion scenarios in our proposed framework.

**Green Light Duration Redistribution (Signal Malfunction)** To evaluate the LLM's ability to see if it can recover from severe traffic signal malfunctions, we intentionally designed two extreme failure scenarios—one targeting the entire North-South (NS and NS-L) approach and another one targeting the entire East-West (EW and EW-L) approach. In these cases, the affected approach was intentionally starved by keeping one or more signals red for an abnormally long duration. For Example, the green signal was withheld for 200 seconds (exceeding two full cycles, where each cycle is 90 seconds), followed by only 4 seconds of green time per subsequent cycle in the north approach, which caused both NS and NS-L lanes to suffer. This setup created an extreme level of congestion in the affected lanes. When LLM is commanded, it responds by generating a corrective signal plan that provides longer green durations based on the highest average occupancy percentage observed in each lane group. It produces four timing plans, one for each of the predefined phase groups: NS, NS-L, EW, and EW-L. Each plan adjusts the green time allocation based on congestion severity. Example: Consider a case where the entire East approach is suffering from a traffic signal malfunction. Then it is noticed that the EW phase group suffers the most congestion. An appropriate prompt is passed to the LLM:

*Traffic jam in EW because of signal failure, estimated congestion: 48%*

In response, the model generates a new Python script. That script outputs a new tllogic.add.xml with proper timing plan where green duration for the EW phase is increased most then EW-L phase is given second priority while durations for the other two phase groups are proportionally reduced—allowing the queue to clear faster.

**Direction Prioritization During Traffic Surges (Event-driven)** In scenarios simulating event-driven surges—such as a burst of vehicles in any of the twelve directional approaches (e.g., South to North, North to East, East to West)—the LLM analyzes congestion by monitoring average phase occupancy levels. Based on the affected phase group (e.g., NS, NS-L, EW, EW-L), it generates a new signal plan with more green time allocated to that phase. For example, if left-turn lanes like N-L (North to East) or S-L (South to West) are congested, an appropriate prompt is passed to the LLM:

*NS-L has Highest Congestion with 50% occupancy*

Then LLM generates python code to provide more green time to the NS-L phase to help clear the buildup more efficiently. This approach allows the excess queue to clear faster from the overloaded direction, while maintaining a balanced situation in other phases. The LLM always generates four updated signal plans—one for each phase group.

**Obstruction-aware Phase Adjustment (Stalled Vehicle)** When a vehicle breaks down in a specific lane, such as the left turn lane, the affected lane becomes temporarily inaccessible, as it creates a severe situation by blocking other vehicles behind it from proceeding. This not only delays other vehicles in that lane but can also lead to system-wide congestion due to its effects across connected

lanes. So LLM uses its thinking power to detect the phases that correspond to the affected lanes, then solves the situation by reducing green time in the affected phases and extending green times in unaffected phases until the vehicle is taken away. After the vehicle is taken away, LLM prioritizes the affected phases, extending green time and reducing it for the unaffected phases. LLM's optimization logic is based on the fact that giving more green time to affected phases is not convenient, as it wastes green time due to vehicles behind the stalled car being unable to proceed. For instance, consider the following scenario where a car breaks down near the east left turn intersection. An appropriate prompt is passed to the LLM:

> *Generate optimized traffic signal plan for a broken car near the west left-turn intersection.*

Model responds by generating a Python script, which in result reduces the green time allocated to that phase (EW-L) and reallocates it to unaffected phases (NS, NS-L, EW) until the vehicle is towed or removed. Once the obstruction is cleared (e.g., the stalled vehicle is towed away), the signal timing is re-adjusted to give priority to the previously blocked phase (EW-L). EW-L phase gets more green time while other phases get less green time.

## 4 EXPERIMENTS

### 4.1 MODEL ARCHITECTURE AND FINE-TUNING

#### 4.1.1 LLaMA 3.2 1B WITH QLoRA

We employ supervised fine-tuning (SFT) of the LLaMA 3.2 1B-Instruct model on a dataset of prompt–code pairs (around 5k samples) to generate SUMO-compatible Python code. To reduce memory usage and computational overhead, we are using the QLoRA framework inspired by (Dettmers et al., 2023), which combines **4-bit quantization of the base model** with **LoRA adapters for parameter-efficient fine-tuning**. Specifically, we use the following materials in our fine-tuning pipeline:

**4-bit Quantization:** When loading the model, we employ a 4-bit compression method (NF4) to downscale the precision of the pretrained model's weights from 16-bit to 4-bit. Double quantization is enabled, which shrinks the quantization constants even more and ensures lower memory overhead. Pretrained weights are approximately normal $\mathcal{N}(0, \sigma^2)$, we then map them to a fixed distribution by normalizing each block of weights to the range $[-1, 1]$. A codebook with $2^k$ levels (here $k = 4$) is then derived from the quantiles of the standard normal distribution:

$$q_i = \tfrac{1}{2}\Big(Q_{\mathcal{N}(0,1)}\Big(\tfrac{i}{2^k+1}\Big) + Q_{\mathcal{N}(0,1)}\Big(\tfrac{i+1}{2^k+1}\Big)\Big), \quad i = 1, \ldots, 2^k,$$

where $Q_{\mathcal{N}(0,1)}$ is the quantile function.

**LoRA Adapters in QLoRA:** Instead of updating the entire 1 B parameters, we add trainable low-rank matrices (rank $r = 8$, scaling $\alpha = 16$) into selected transformer layers: q_proj, k_proj, v_proj, o_proj, gate_proj, up_proj, and down_proj. Given a frozen quantized weight matrix $W_q$, we define $\Delta W$ as the low-rank trainable weight update introduced by LoRA:

$$\Delta W = \tfrac{\alpha}{r} AB,$$

so that the effective weight used during training is

$$W_{\text{eff}} = W_q + \Delta W.$$

Here, $A$ and $B$ are the only trainable parameters, while $W_q$ remains frozen. Thus, QLoRA achieves full-model fine-tuning behavior while only training a few million parameters while keeping the knowledge of the pretrained model intact.

**Optimizer:** We are using 8-bit paged AdamW optimizer, which minimizes memory usage without damaging the training stability. Gradient updates are applied exclusively to LoRA adapters . Gradient accumulation is used to handle larger batch sizes on limited hardware. **Training Configuration:** The model is trained for 4 epochs with a batch size of 4, gradient accumulation steps of 4, and learning rate of $2 \times 10^{-4}$. Gradient checkpointing is enabled to reduce memory usage. The dataset is split into 80% for training and 20% for validation.

## 4.2 SIMULATION AND EVALUATION

### 4.2.1 SUMO INTEGRATION

The output from the fine-tuned LLM consists of Python code that outputs XML configuration files in the SUMO traffic simulator. Depending on the type of scenario, the generated code creates one of the following:

**route.rou.xml** — Defines vehicle routes, departure times, speed, vehicle types, and traffic volumes. This is used for scene generation tasks such as adding vehicles, simulating stalled vehicles, or generating heavy or medium traffic flows.

**tllogic.add.xml** — Defines custom traffic signal logic. This is created when the LLM is prompted to optimize traffic signal timings in response to congestion scenarios such as signal malfunctions, sudden bursts of traffic, or stalled vehicles.

LLM generates a Python script when it is called. Then this code produces these XML files. Once these XML files are generated, they are loaded into the SUMO to run the simulation to copy real-world traffic conditions. The effectiveness of the LLM-generated control logic is determined by using two well-known traffic performance metrics:

**Average Travel Time (ATT)**: The average time all the vehicles take to travel from their source to their destination.

**Average Waiting Time (AWT)**: The average time all the vehicles wait at intersections.

### 4.2.2 EVALUATION METRICS UNDER STALLED VEHICLE SCENARIO

To evaluate the outcome of the LLM-generated traffic signal plans, we selected two representative scenarios out of four: one with a broken vehicle near the west intersection and another near the north intersection. Although we have respective scenarios for the east and south approaches, we are focusing on the west and north cases for illustrative purposes. We observed that when the overall vehicle count is low, the differences in traffic metrics before and after applying the LLM-optimized logic are minimal. However, when the traffic flows are very high, lanes remain occupied for some period of time, and the optimized signal plans lead to significant improvements in performance metrics. For both scenarios, the west left turn and the east left turn, we compare ATT and AWT before and after deploying the LLM-generated traffic light adjustments. The results are summarized in Table 1.

| Lane | Metric | Before LLM | After LLM |
|------|--------|------------|-----------|
| W-L | ATT | 932.88 | 721.27 |
| W-L | AWT | 63.59 | 48.08 |
| N-L | ATT | 954.86 | 640.98 |
| N-L | AWT | 63.94 | 43.05 |

Table 1: Metrics under stalled vehicle scenario.

### 4.2.3 EVALUATION METRICS UNDER SUDDEN BURST SCENARIO

To evaluate the effectiveness of the LLM in mitigating congestion caused by sudden bursts of traffic, we conducted high-density traffic simulations from four directions — North to East, North to South, East to West, and East to South — belonging to the NS, NS-L, EW, and EW-L signal phase groups, respectively. Although our system was trained and tested on all 12 directional flows at the intersection, these four directions were selected for detailed analysis. We observed that EW-L routes from East to South and NS-L routes from North to East experienced higher congestion levels and longer delays. These directional approaches are left turn lanes. So vehicles in left turn lanes take more time to clear the intersection than straight lane movements. The delay and wait time in these lanes is always higher than in other lanes.

| Phase | Metric | Before LLM | After LLM |
|-------|--------|------------|-----------|
| NS | ATT | 93.97 | 38.09 |
| NS | AWT | 60.89 | 14.04 |
| NS-L | ATT | 1024.42 | 143.39 |
| NS-L | AWT | 81.82 | 24.47 |
| EW | ATT | 96.13 | 47.55 |
| EW | AWT | 61.76 | 20.82 |
| EW-L | ATT | 397.92 | 41.84 |
| EW-L | AWT | 81.53 | 12.39 |

Table 2: Comparison of ATT and AWT Across four phases before and after LLM optimization.

#### 4.2.4 EVALUATION METRICS UNDER FAULTY TRAFFIC LIGHT SCENARIO

This extreme traffic signal malfunctioning setup was designed to starve one directional approach for an abnormally longer period, barely giving any green light time to the vehicles to clear the lanes. Thus, a severe level of congestion happens at the intersection. The table below demonstrates how this extreme malfunction led to a dramatic increase in average travel time (ATT) and average waiting time (AWT) before intervention. After applying the LLM-generated signal plans, the system significantly reduced congestion by reallocating green durations based on phase-level occupancy. The difference in ATT and AWT before and after LLM is being used to generate code to optimize traffic flow is much higher, which again reflects the extreme imbalance of the traffic signal configuration in the intersection. Table 3 reports ATT and AWT before and after applying LLM-generated code for signal optimization under extreme congestion conditions.

| Phase | Metric | Before LLM | After LLM |
|-------|--------|------------|-----------|
| NS or NS-L | ATT | 2146.73 | 105.3 |
| NS or NS-L | AWT | 75.88 | 41.8 |
| EW or EW-L | ATT | 2261.7 | 99.7 |
| EW or EW-L | AWT | 80.62 | 38.7 |

Table 3: Metrics under faulty traffic light scenario.

## 5 CONCLUSION

Our framework uses a large language model to generate and simulate both normal and adversarial traffic scenarios. Most existing frameworks rely on synthetic data, while a few use real-time data. Some of them use LLMs only to output which phase to activate for traffic signal control. In contrast, our framework derives its dataset from real-world traffic videos, tracks each vehicle along with its direction of movement, speed, and departure time, and incorporates all this information into SUMO simulation, making the simulation more realistic. The LLM then generates code to provide appropriate traffic signal plans to manage congestion. Existing frameworks do not combine all these components into a unified model. From the experimental results, we observe that evaluation metrics like ATT and AWT improve significantly after the LLM is used to manage traffic signal control.

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

## A  DESIGN SPACE COMPARISON

| Paper | Data | Task | Propose Signal Plan | Metrics used for Prompt |
|-------|------|------|---------------------|-------------------------|
| ChatSUMO | Synthetic | Scenario Generation | No | None |
| SUMO–MCP | Synthetic | Scenario Generation | No | None |
| LLM-Assisted Light | Real+Synthetic | Signal Control | Yes | Current phase; Current occupancy |
| CoLLMLight | Synthetic | Signal Control | Yes | Current phase; Queue length |
| **Ours** | Real+Synthetic (video feed) | Scenario and Signal Generation | Yes | Max-occupancy phase; Max avg. occupancy |

Table 4: Comparison with other LLM-based papers

## B  ADDITIONAL RESULTS

### B.1  PROMPT–RESPONSE EXAMPLES IN TRAFFIC SCENARIO AND CONTROL GENERATION

**Normal Traffic Flow**

> **👤 Prompt**
>
> add 3 cars, from west to south, east to north, west to east, departing at 92.79, 95.46, 95.69 seconds

> **🤖 Answer**
>
> ```python
> import xml.etree.ElementTree as ET
>
> XML_PATH = r"C:\path\to\routes.rou.xml"
> ```

```
def next_ids(root, n):
    base = max((int(v.get("id").lstrip("veh")) for v in root.findall
    ("vehicle")),
               default=-1) + 1
    return [f"veh{base+i}" for i in range(n)]

def add_and_sort(routes, departs):
    tree = ET.parse(XML_PATH)
    root = tree.getroot()
    ids = next_ids(root, len(routes))
    for vid, r, d in zip(ids, routes, departs):
        root.append(ET.Element("vehicle", {
            "id": vid,
            "route": r,
            "depart": f"{d:.2f}".rstrip("0").rstrip(".")
        }))
    vs = sorted(root.findall("vehicle"), key=lambda v: float(v.get("
    depart")))
    for v in vs:
        root.remove(v)
        root.append(v)
    tree.write(XML_PATH, encoding="utf-8", xml_declaration=True)

if __name__ == '__main__':
    ROUTES  = ['route_west_to_south',
               'route_east_to_north',
               'route_west_to_east',
               ]
    DEPARTS = [92.79, 95.46, 95.69]
    add_and_sort(ROUTES, DEPARTS)
```

**Heavy/Medium Traffic Flow**

👤 **Prompt**

I want to see heavy traffic flow of 259 vehicles.

🤖 **Answer**

```
import random, xml.etree.ElementTree as ET
from pathlib import Path
OUT     = Path(r"route.rou.xml")
N_VEH   = 259
SIM_END = 1000
SEED    = 42
random.seed(SEED)
ROUTES = [('route_north_to_east', 'north_to_center center_to_east'),
     ('route_north_to_south', 'north_to_center center_to_south'), ('
    route_north_to_west', 'north_to_center center_to_west'), ('
    route_east_to_west', 'east_to_center center_to_west'), ('
    route_east_to_south', 'east_to_center center_to_south'), ('
    route_east_to_north', 'east_to_center center_to_north'), ('
    route_south_to_west','south_to_center center_to_west'), ('
    route_south_to_north','south_to_center center_to_north'), ('
    route_south_to_east','south_to_center center_to_east'), ('
    route_west_to_south', 'west_to_center center_to_south'), ('
    route_west_to_north', 'west_to_center center_to_north'), ('
    route_west_to_east', 'west_to_center center_to_east')]
root = ET.Element("routes")
```

```
648
649    for r_id, edges in ROUTES:
650        ET.SubElement(root, "route", id=r_id, edges=edges)
651    vehicles = []
652    for i in range(N_VEH):
653        depart  = round(random.uniform(0, SIM_END), 2)
654        routeID = random.choice(ROUTES)[0]
655        v = ET.Element("vehicle")
656        v.set("id",      f"veh{i}")
657        v.set("route",   routeID)
        v.set("depart", f"{depart}")
658        vehicles.append(v)
659    for v in sorted(vehicles, key=lambda e: float(e.get("depart"))):
660        root.append(v)
661    ET.ElementTree(root).write(OUT, encoding="utf-8", xml_declaration=
        True)
662    print(f" wrote {OUT} with {N_VEH} vehicles (0{SIM_END} s)")
663
```

**Sudden Vehicle Breakdown**

### 👤 Prompt

A car left at 5.48 s stalls in the west left-turn pocket, backing up traffic.

### 🤖 Answer

```
import xml.etree.ElementTree as ET, traci
from pathlib import Path
from sumolib import checkBinary
CFG_FILE, ROUTE_FILE = r"sumo_config.sumocfg", Path(r"route.rou.xml"
    )
BROKEN_ROUTE, BROKEN_LANE = "route_west_to_north", "west_to_center_2
    "
DEPART, STALL, END = 5.48, 800, 2300
root = ET.parse(ROUTE_FILE).getroot()
for n in root.findall('vehicle[@id="broken"]'):
    root.remove(n)
v = ET.SubElement(root, "vehicle",
                  id="broken", route=BROKEN_ROUTE, depart=str(DEPART
    ),
                  color="255,120,0")
ET.SubElement(v, "stop", lane=BROKEN_LANE, pos="82", duration="
    999999")
others = [e for e in root if e.tag!= "vehicle"]
vehicles = sorted(root.findall("vehicle"),
                  key=lambda x: float(x.get('depart', '0')))
root[:] = others + vehicles
ET.ElementTree(root).write(ROUTE_FILE, encoding="utf-8",
    xml_declaration=True)
cmd = [checkBinary("sumo-gui"), "-c", CFG_FILE,
    "--begin", "0", "--end", str(END),
    "--collision.action", "none",
    "--time-to-teleport", "-1"]
traci.start(cmd)
tow_time = DEPART + STALL
while True:
    traci.simulationStep()
    t = traci.simulation.getTime()
    if t >= tow_time and "broken" in traci.vehicle.getIDList():
        traci.vehicle.remove("broken")
```

```
    if t >= END:
        break
traci.close()
```

**Traffic Signal Malfunctioning**

**👤 Prompt**

Traffic jam in NS-L because of signal failure, estimated congestion: 48%

**🤖 Answer**

```
import xml.etree.ElementTree as ET
additional = ET.Element("additional")
tl_logic = ET.SubElement(additional, "tlLogic", {
    "id": "center",
    "type": "static",
    "programID": "sudden_ns-l",
    "offset": "0"
})
ET.SubElement(tl_logic, "phase", {"duration": "25", "state": "
    rrrGGrrrrrrrrGGrrrrr"})
ET.SubElement(tl_logic, "phase", {"duration": "5", "state": "
    rrryyrrrrrrrryyrrrrr"})
ET.SubElement(tl_logic, "phase", {"duration": "30", "state": "
    GGGGGrrrrrGGGGGrrrrr"})
ET.SubElement(tl_logic, "phase", {"duration": "5", "state": "
    yyyyyrrrrryyyyyrrrrr"})
ET.SubElement(tl_logic, "phase", {"duration": "20", "state": "
    rrrrrGGGGGrrrrrGGGGG"})
ET.SubElement(tl_logic, "phase", {"duration": "5", "state": "
    rrrrryyyyyrrrrryyyyy"})
tree = ET.ElementTree(additional)
tree.write("add_ns-l.xml", encoding="utf-8", xml_declaration=True)
```

**Sudden Burst of Vehicles**

**👤 Prompt**

EW has Highest Congestion with 50% occupancy

**🤖 Answer**

```
import xml.etree.ElementTree as ET
root = ET.Element("additional")
tl = ET.SubElement(root, "tlLogic", id="center", type="static",
    programID="solution_faulty_lights_ew", offset="0")
ET.SubElement(tl, "phase", duration="45", state="
    rrrrrGGGGGrrrrrGGGGG")
ET.SubElement(tl, "phase", duration="10", state="
    rrrrryyyyyrrrrryyyyy")
ET.SubElement(tl, "phase", duration="25", state="
    GGGGGrrrrrGGGGGrrrrr")
ET.SubElement(tl, "phase", duration="10", state="
    yyyyyrrrrryyyyyrrrrr")
```

```
tree = ET.ElementTree(root)
tree.write("tlLogic_output.add.xml", encoding="utf-8",
    xml_declaration=True)
```

**Stalled Vehicle near left lane in intersection**

### 👤 Prompt

Generate optimized traffic signal plan for a broken car near the east left-turn intersection.
Scenario 10

### 🤖 Answer

```python
import xml.etree.ElementTree as ET
root = ET.Element("additional")
broken_logic = ET.SubElement(root, "tlLogic", id="center", type="
    static", programID="broken_eleft", offset="0")
for duration, state in [
    ("25", "rrrrrGGGrrrrrrGGGGG"), ("5", "rrrrryyyrrrrrrryyyyy"), (
    "1", "rrrrrGGGGGrrrrrGGGGG"), ("4", "rrrrryyyyyrrrrryyyyy"), ("
    50", "GGGGGrrrrrGGGGGrrrrr"), ("5", "yyyyyrrrrryyyyyrrrrr")
]:
    ET.SubElement(broken_logic, "phase", duration=duration.strip('"'
    ), state=state.strip('"'))
faulty_logic = ET.SubElement(root, "tlLogic", id="center", type="
    static", programID="faulty_EL", offset="0")
for duration, state in [
    ("50", "rrrrrGGGGGrrrrrGGGGG"), ("10", "rrrrryyyyyrrrrryyyyy"),
    ("20", "GGGGGrrrrrGGGGGrrrrr"), ("10", "yyyyyrrrrryyyyyrrrrr")
]:
    ET.SubElement(faulty_logic, "phase", duration=duration.strip('"'
    ), state=state.strip('"'))
tree = ET.ElementTree(root)
tree.write("generated_broken_eleft.add.xml", encoding="utf-8",
    xml_declaration=True)
```

## B.2 VISUALIZATION OF THE RESULTS

**Stalled Vehicle Scenario** At $t = 357$ s, the stalled vehicle (orange car) in the `EW-L` lane is still present, so it cannot discharge. The LLM-generated Python script responds by writing an updated `tlLogic.add.xml` that temporarily reduces green allocated to `EW-L` and reallocates it to `NS`, `NS-L`, and `EW` to keep the intersection flowing. This behavior prioritizes phases that can actually serve demand while the obstruction persists. Once the vehicle is removed, the timing plan rebalances to restore priority to `EW-L`, granting it additional green while proportionally shortening the others. We therefore compare outcomes at $t = 357$ s (vehicle not yet towed) with and without the LLM plan in figure 3

**Traffic Signal Malfunction** At $t = 577$ s, the northbound approach is heavily congested without the LLM plan, whereas with the LLM plan at the same timestamp the northbound vehicle count drops markedly in figure 4. Given these congestion inputs, the LLM generates Python that writes `tlLogic.add.xml` to extend green for `NS` (then `NS-L`) while proportionally shortening the other phases, yielding the observed reduction.

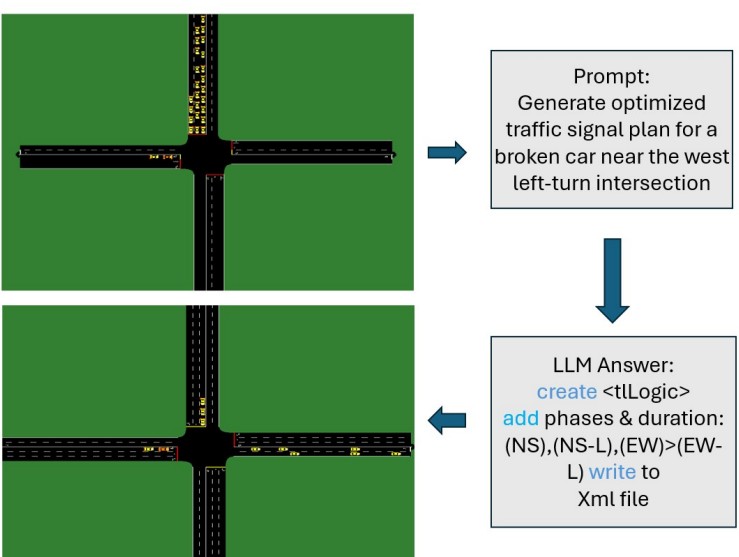

Figure 3: comparison without LLM vs. with LLM-generated timing plan in broken car scenario.

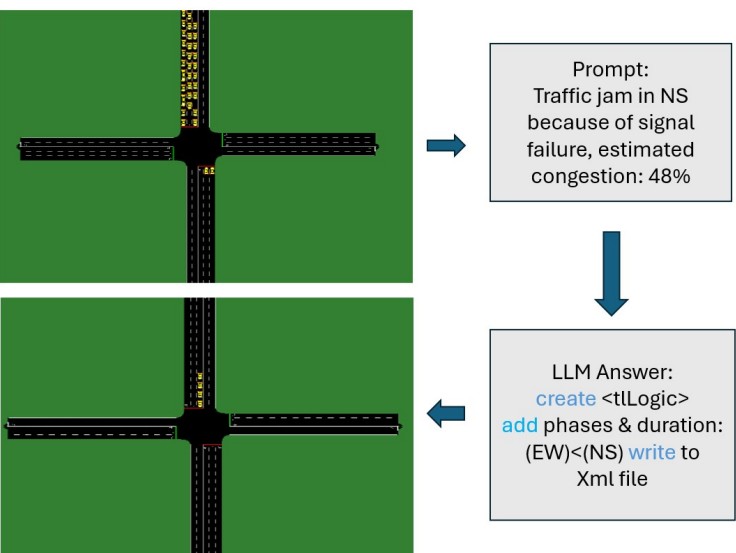

Figure 4: Traffic signal malfunctioning: comparison without LLM vs. with LLM-generated timing plan

## C    DATASET AND ALGORITHM

### C.1    DATASET COLLECTION

We utilized a real-world traffic dataset collected from the Bellevue intersection in Washington. City of Bellevue, Washington (2017). Vehicle trajectories were extracted using YOLO object detection and the DeepSORT tracking algorithm, which allowed us to determine the number of vehicles entering from each approach, their respective movements, average speed, and departure times. Subsequently, these trajectories were integrated into the SUMO traffic simulator to follow realistic traffic flow conditions.

The dataset spans eight days and was segmented into six temporal intervals: 1-minute, 2-minute, 3-minute, 4-minute, 5-minute, and 6-minute durations. For sudden bursts of vehicles, we used

1-minute and 2-minute duration videos. For faulty traffic signal scenarios, we used 2-minute, 3-minute, 4-minute, 5-minute, and 6-minute videos. We trained on 6 days of data, then tested on 2 days of unseen data. Based on this, we introduced a variety of congestion-inducing scenarios to simulate more complex and realistic urban conditions. These included:

1. Placement of stalled vehicles in the left lane from north/south/east/west directions to simulate road obstructions,

2. Extreme malfunctioning of the traffic signal, which results in an unusually longer red phase in one direction, and

3. Sudden bursts of vehicles during peak hours to emulate event-driven surges.

Following the integration of these scenarios, we measured the average occupancy percentage for each signal phase—defined as the proportion of the lane occupied by vehicles. The average occupancy percentage served as key input for the LLM to assess congestion levels and generate optimized traffic signal control plans.

## C.2  ALGORITHM OF THE SYSTEM

The workflow monitors traffic phases, computes occupancies, and detects congestion types (stalled vehicle, malfunction, or burst). It then queries the LLM with occupancy data and congestion type to generate updated signal plans. Finally, SUMO is rerun with the new plans and performance metrics are compared.

---

**Algorithm 1** LLM-Guided Traffic Signal Optimization Workflow

---

1: **Initialize** SUMO simulation
2: **while** simulation not finished **do**
3:     Map each lane to its phase group $\{\text{NS}, \text{NS-L}, \text{EW}, \text{EW-L}\}$
4:     Compute per-phase average occupancy $\bar{O}_p$
5:     $p^\star \leftarrow \arg\max_p \bar{O}_p$
6:     **if** stalled vehicle detected **then**
7:         Mark congestion type $\leftarrow$ STALLED VEHICLE
8:         Record towing time, vehicle departure time, affected lane
9:     **else if** signal malfunction detected **then**
10:         Mark congestion type $\leftarrow$ SIGNAL MALFUNCTION
11:     **else**
12:         Mark congestion type $\leftarrow$ SUDDEN BURST
13:     **end if**
14: **end while**
15: **Call** LLM with $(\{\bar{O}_p\}, \text{congestion type})$
16: **Receive** Python code that generates new signal plans
17: Run SUMO with new signal plans
18: Record and compare metrics (ATT, AWT) before and after LLM

---

Algorithm 1 illustrates how the system handles each congestion event.

