# OpenReview forum: "Generative Traffic Simulations"
_ICLR.cc/2026/Conference — Submitted to ICLR 2026_

### Official Review · Reviewer_bpCc · 2025-10-21

**Soundness:** 2
**Presentation:** 1
**Contribution:** 1
**Rating:** 2
**Confidence:** 5

**Summary:**

This paper presents a cascaded AI system that integrates a deep neural network with a large language model (LLM) to generate both routine and edge-case traffic scenarios from real-world video feeds and optimize offline traffic signal plans in SUMO, significantly improving travel efficiency and reducing delays.

**Strengths:**

1. This paper attempts to leverage LLMs to address complex traffic scenarios that require human-like understanding and reasoning.
2. This paper presents an complete pipeline within the SUMO simulator, covering real-world data extraction, synthetic traffic scenario generation, signal timing optimization, and handling of emergency or disruptive events.

**Weaknesses:**

1. The overall writing quality is low, particularly in terms of logical structure, problem framing, articulation of technical novelty, and clarity of presentation. Specific issues are detailed below.
2. The title, “Generative Traffic Simulations,” is overly broad and generic. It describes an entire research area rather than a specific paper, and it poorly reflects the actual scope and focus of this work.
3. The introduction extensively surveys existing traffic simulators (e.g., SUMO, CARLA) and signal control methods (rule-based, RL, LLM-based), but fails to clearly define the specific problem the paper aims to solve or the unique challenge it addresses. The stated contributions read more like a list of components than a coherent research narrative.
4. The core methodology, using prompts to drive an LLM to generate SUMO-compatible XML/Python code for scenarios and signal plans, is essentially a code-generation task. While LLM-based code generation is well-studied, the paper neither demonstrates a significant technical advance over existing approaches nor includes comparisons with relevant baselines in this space.
5. The work touches on many types of traffic scenarios (normal flow, vehicle breakdowns, signal malfunctions, traffic bursts), but lacks focus. As a result, the analysis remains superficial, and the paper does not convincingly demonstrate which specific traffic control problems are uniquely or substantially better addressed by using LLMs.

**Questions:**

1. What is the precise technical challenge this paper seeks to solve? Which aspects are already addressable by existing methods, and where exactly lies the unresolved gap that this work fills?
2. What is the complete technical architecture of the proposed system? A clear, unified framework diagram or formal description is missing.
3. What baseline methods are used for comparison in the experiments? The evaluation should include comparisons against traditional signal control strategies and recent LLM-based traffic control approaches.
4. Beyond generating SUMO simulation configurations, can the proposed LLM-driven approach meaningfully improve traditional traffic signal optimization pipelines? If so, how?

---

### Official Review · Reviewer_5gCx · 2025-10-25

**Soundness:** 2
**Presentation:** 2
**Contribution:** 1
**Rating:** 2
**Confidence:** 3

**Summary:**

The paper proposes to use an LLM based code-generator to synthesize traffic scenarios from natural-language prompts, and produce offline, phase-based signal timing plans that are re-run in SUMO to reduce congestion. Experiments show sizable gains in Average Travel Time and Average Waiting Time across the three stressors

**Strengths:**

-paper is overall clear

**Weaknesses:**

-the methodological contribution is incremental as using LLM to generate code for traffic scenarios and traffic plan have been studied in previous work.

-evaluation is limited (only one junction) and no baseline controllers have been compared

**Questions:**

-How does the method compare against other controllers?

---

### Official Review · Reviewer_CMm9 · 2025-10-28

**Soundness:** 2
**Presentation:** 1
**Contribution:** 2
**Rating:** 2
**Confidence:** 4

**Summary:**

This paper proposes a method for SUMO scenario generation and traffic signal control using a large language model (LLM). The approach involves fine-tuning a LLaMA-3.2-1B model to automatically generate SUMO configuration files. Experiments conducted on a simple intersection scenario demonstrate the potential of LLMs in both generating scenario files and optimizing traffic light configurations. However, the experimental setup is relatively simple, and the study lacks comparative experiments.

**Strengths:**

This paper utilizes a large language model (LLM) to address the problems of scenario generation and signal control within a unified framework. The proposed method can automatically generate novel scenarios and control strategies, which holds significant potential to benefit model training in the field of signal control.

**Weaknesses:**

The experimental setup, which uses only a single intersection, is too simplistic to highlight the advantages of the proposed method. The authors also do not explain the specific contributions that improve the model's performance in either generating SUMO configurations or formulating signal control policies.

**Questions:**

- The abstract claims that the proposed method generates scenarios using real-world traffic data. However, this aspect is not addressed or substantiated in the main body of the paper. The authors should either include the relevant methodology and results or revise the claim in the abstract.
- The paper does not include comparative experiments against baseline or state-of-the-art methods for scene generation and signal control. Without such a benchmark, it is difficult to assess the superiority and effectiveness of the proposed approach.
- The experimental design appears to be too simplistic and does not fully align with the ambitious vision outlined in the abstract and introduction. The experiments should be redesigned to better reflect the complexity of the problem the authors aim to solve.

---

### Official Review · Reviewer_WKWk · 2025-10-28

**Soundness:** 3
**Presentation:** 3
**Contribution:** 3
**Rating:** 4
**Confidence:** 4

**Summary:**

The paper proposes a cascaded AI system for generative traffic simulation and signal plan synthesis centered on a single four-way intersection in SUMO. The system couples (i) a fine-tuned LLaMA-3.2-1B (via QLoRA) that generates Python code to write SUMO XML for scenarios and static signal plans, with (ii) a lane/phase occupancy analysis pipeline that identifies the most congested phase and prompts the LLM to re-allocate green times. Three scenario families are considered: normal traffic, edge cases via stalled vehicles, and event-driven bursts; an additional signal-malfunction stress test is used. Effectiveness is reported through Average Travel Time (ATT) and Average Waiting Time (AWT) before/after applying LLM-generated timing plans, with large relative improvements in the presented cases.

**Strengths:**

1. The paper’s core contribution—linking LLM-based code generation with data-driven traffic phase optimization—is both novel and coherent.
2. The proposed pipeline is complete, transparent, and reproducible: from the QLoRA-based fine-tuning of LLaMA-3.2-1B, through SUMO interface logic, to congestion identification and re-prompting.

**Weaknesses:**

1. No controlled experiments versus Webster, Max-Pressure/Modified-Pressure, PressLight/IntelliLight, or recent LLM controllers on identical demand patterns. Without these, improvements could reflect scenario choices rather than superior policy.
2. The method doesn’t address network-level spillbacks, offsets, or progression; and because it is offline, it cannot react within a run to evolving queues or detector noise.
3. The paper lacks error statistics (invalid XML, conflicting signal states), runtime overhead for generation/IO, and guardrails (schema validation, conflict matrices).

**Questions:**

1. How sensitive are outcomes to: (a) prompt phrasing, (b) the max-occupancy rule vs. alternatives (e.g., pressure, queue length, delay), and (c) the congestion-type classifier? Provide ablation curves.
2. What fraction of LLM outputs require manual fixes? Do you use XML schema validation or conflict matrices to prevent unsafe greens? Provide error rates, examples, and mitigation.
3. Can the method handle multi-intersection corridors (with offsets and progression) or network-wide cases? If not, what changes are needed (e.g., hierarchical prompting, decentralized plans)?

---

### Meta-Review · Area_Chair_PB6J · 2025-12-16

**Summary:**

The submission received 2,2,2,4. There is a strong consensus of rejection. No rebuttals nor responses are provided by the authors. Thus, a rejection is recommended.

**Reviewer Scores:**

NA

---

### Decision · Program_Chairs · 2026-01-26

Reject